# Computation identifies structural features that govern neuronal firing properties in slowly adapting touch receptors

Daine R Lesniak[1†], Kara L Marshall[2†], Scott A Wellnitz[3‡], Blair A Jenkins[2,4], Yoshichika Baba[2], Matthew N Rasband[3], Gregory J Gerling[1], Ellen A Lumpkin[2,5*]

[1]Department of Systems and Information Engineering, University of Virginia, Charlottesville, United States; [2]Department of Dermatology, Columbia University, New York, United States; [3]Department of Neuroscience, Baylor College of Medicine, Houston, United States; [4]Medical Scientist Training Program, Columbia University, New York, United States; [5]Department of Physiology and Cellular Biophysics, Columbia University, New York, United States

**Abstract** Touch is encoded by cutaneous sensory neurons with diverse morphologies and physiological outputs. How neuronal architecture influences response properties is unknown. To elucidate the origin of firing patterns in branched mechanoreceptors, we combined neuroanatomy, electrophysiology and computation to analyze mouse slowly adapting type I (SAI) afferents. These vertebrate touch receptors, which innervate Merkel cells, encode shape and texture. SAI afferents displayed a high degree of variability in touch-evoked firing and peripheral anatomy. The functional consequence of differences in anatomical architecture was tested by constructing network models representing sequential steps of mechanosensory encoding: skin displacement at touch receptors, mechanotransduction and action-potential initiation. A systematic survey of arbor configurations predicted that the arrangement of mechanotransduction sites at heminodes is a key structural feature that accounts in part for an afferent's firing properties. These findings identify an anatomical correlate and plausible mechanism to explain the driver effect first described by Adrian and Zotterman.

*For correspondence: eal2166@columbia.edu

†These authors contributed equally to this work

‡Present address: The Gladstone Institutes, San Francisco, United States

**Competing interests:** The authors declare that no competing interests exist.

## Introduction

A diverse array of touch receptors allows animals to discern object shapes, to explore surface textures and to detect forces impinging upon the skin. In mammals, distinct classes of mechanosensory afferents are tuned to extract specific features of a tactile stimulus and then to encode them as trains of action potentials, or spikes, with unique firing properties (*Johnson, 2001*). A common feature of mechanosensory neurons is specialized anatomical structures, termed end organs, that shape their neuronal outputs (*Chalfie, 2009*). For example, recent studies show that mouse hair follicles are innervated by at least three molecularly (*Li et al., 2011*) and 10 anatomically (*Wu et al., 2012*) distinct types of cutaneous afferents. A key unanswered question is: how does a tactile afferent's peripheral architecture govern its neuronal response to touch stimuli?

Due to their unusual architecture, somatosensory neurons do not initiate action potentials at axon initial segments, as do neurons of the central nervous system. Instead, sensory stimuli act at peripheral terminals to produce receptor potentials, which locally sum to trigger spikes that travel to central terminals up to 1 m away. For myelinated tactile afferents, a landmark study of Pacinian corpuscles established that spikes initiate at the heminode, the most distal node of Ranvier (*Loewenstein and Rathkamp, 1958*). A Pacinian corpuscle is innervated by an un-branched afferent; however, most tactile end organs comprise branching afferents with multiple sites of sensory transduction.

**eLife digest** Sensory receptors in the skin supply us with information about objects in the world around us, including their shape and texture. These receptors also detect pressure, temperature, and pain, enabling us to respond appropriately to stimuli that could be potentially harmful.

The activation of a touch receptor—for example, due to the movement of a hair—causes ions to flow into the cell, changing the electric charge inside it. When the charge exceeds a threshold value, the cell fires action potentials, which travel along its axon to the central nervous system. The patterns of these action potentials from a population of touch receptors carry all the information about a touch stimulus to the brain. Different types of sensory receptors have unique anatomical structures and distinct signaling patterns; however, little is known about how the structures of sensory receptors influence action potential firing.

Now Lesniak and Marshall et al. have revealed that structure determines function in a type of mammalian touch receptor called the slowly adapting type I receptor, which is concentrated in fingertips and other areas of high tactile acuity. With the aid of high-resolution microscopy, the complex branching structure of the receptor and its network of nerve endings were mapped in three dimensions. Experiments revealed highly variable structures and firing patterns between individual touch receptors, and computational modeling showed that changing either the number or the arrangement of receptor endings influenced the neuron's firing properties.

This is the first computational model that captures touch encoding by combining skin properties, sensory transduction, and spike initiation. As well as providing new information on how structure permits function, this work opens up new possibilities for exploring how the skin maintains its sensory capabilities during routine maintenance and after injury.

The question of how spike trains arise in branched sensory neurons has fascinated neurobiologists since Adrian and Zotterman (*Adrian and Zotterman, 1926a*). In the simplest configuration, which is observed in crustacean stretch receptors and frog muscle spindles, receptor potentials from all branches integrate at a single spike initiation zone (*Adrian and Zotterman, 1926b*). As stimulus intensity increases, additional transduction sites are recruited, producing larger receptor potentials to reach spike threshold. Thus, this configuration results in firing rates proportional to the number of transduction sites recruited. Alternatively, sensory afferents can have multiple spike initiation zones, each driven by inputs from one or a few branches (*Horch et al., 1974*). Support for this model comes from studies of mammalian muscle spindles and tendon organs, which have multiple myelinated branches and heminodes where spikes might initiate (*Fukami, 1980*; *Quick et al., 1980*; *Banks et al., 1997*). When a stimulus excites multiple branches, a spike produced by one zone is thought to propagate antidromically into other branches, activating other spike initiation zones and thereby suppressing firing during their refractory period. As a consequence of this resetting mechanism, the spike initiation zone with the highest firing rate is thought to act as a driver for firing in the afferent as a whole. Electrophysiological studies provide strong support for this model (*Lindblom and Tapper, 1966*; *Horch et al., 1974*; *Fukami, 1980*; *Peng et al., 1999*); however, the structural principles that govern spike initiation and integration in mammalian tactile afferents are unknown.

To elucidate the origin of firing patterns in branched tactile receptors, we examined slowly adapting type I (SAI) afferents in mouse skin. These mechanoreceptors localize to skin regions specialized for high tactile acuity, including fingertips, whisker follicles and touch domes. SAI afferents represent fine spatial details with high fidelity; therefore, they are thought to encode object features such as edges and curvature (*Johnson, 2001*). The SAI afferent's end organ is a cluster of Merkel cell-neurite complexes, which are required to produce canonical SAI firing patterns in mouse touch-dome afferents (*Maricich et al., 2009*). Because the essential processes of mechanotransduction and spike initiation occur in tactile end organs, we analyzed the impact of end-organ architecture on touch-evoked responses. As it is not yet possible to directly record from tactile end organs embedded in mammalian skin, we employed a combined experimental and computational modeling approach to identify simple structural principles that can account for the SAI afferent's mechanosensory coding properties.

## Results

### Quantitative morphometric analysis of mouse SAI afferents

SAI afferents are myelinated Aβ afferents that innervate Merkel cells located in the epidermis. Although dermal segments are thickly myelinated, SAI afferents lose their myelin sheaths just below the dermal–epidermal junction. Unmyelinated branches, which are here termed 'neurites', then traverse the basal lamina to contact Merkel cells (*Figure 1A*; *Iggo and Muir, 1969*). To identify structural domains in mouse SAI afferents, we first sought to localize nodes of Ranvier, which are sites of spike integration and propagation, as well as heminodes, which are the anatomical substrates of spike initiation.

We surveyed conserved node proteins in cryosections of adult mouse hairy skin (8–10 weeks of age). We identified SAI afferents by their immunoreactivity to Neurofilament H (NFH; a myelinated-neuron marker) and by their contacts with Keratin-8-positive Merkel cells in touch domes, which are specialized skin regions that surround tylotrich (guard) hairs (*Figure 1B,C*). Myelin Basic Protein (MBP) antibodies were used to visualize myelin end points and gaps, which are the sites of heminodes and nodes of Ranvier, respectively (*Figure 1B,C*). We identified intense, punctate immunoreactivity for the voltage-activated sodium channel $Na_V1.6$ at myelin end points and myelin gaps in cutaneous afferents. In SAI afferents, $Na_V1.6$ puncta localized to 93% of observed myelin end points (N = 28/30, *Figure 1B*) and 100% of myelin gaps (N = 9/9, *Figure 1C*). These data demonstrate that nodes and heminodes in SAI afferents can be reliably identified by visualizing either MBP or $Na_V1.6$. Moreover, they identify $Na_V1.6$ as a principal node component in these cutaneous afferents. We did not observe immunoreactivity against voltage-activated sodium or potassium channels in unmyelinated neurites juxtaposed to Merkel cells (N = 201 Merkel cell-neurite complexes), although it is possible that these channels are present at levels below detection threshold. Based on the strong enrichment of $Na_V1.6$ at heminodes, we infer that spikes likely initiate at these sites, as they do in Pacinian corpuscles (*Loewenstein and Rathkamp, 1958*), rather than initiating in SAI-afferent terminals.

We next sought to quantify the arrangement of afferent branches, nodes and Merkel cells in complete tactile end organs. We employed confocal microscopy and whole-mount skin immunostaining to visualize the entirety of the SAI afferent's end organ (*Figure 1D–F*; *Li et al., 2011*). Myelin end points were capped with $Na_V1.6$-positive heminodes (*Figure 1F*). Unmyelinated neurites that extended from these heminodes branched to contact Merkel cells. In myelinated branches, nodes of Ranvier localized to myelin gaps at every branch point and along un-branching afferent lengths (*Figure 1D,E*). These reconstructions demonstrate that SAI afferents have complex axonal arbors with extensive branching and multiple heminodes and nodes of Ranvier. Thus, we conclude that spikes have the potential to initiate at multiple domains and then to integrate downstream at branch-point nodes in the arbor.

We next traced SAI afferents in three dimensions to quantify structural parameters (*Figure 1D'–F'*). Distributions of nodes identified by MBP and $Na_V1.6$ were indistinguishable, so datasets were pooled for quantitative analysis (*Figure 1G–I*). In 83% of touch domes surveyed, Merkel cells were innervated by branches of a single SAI afferent (N = 18, *Figure 1D–D',F–F'*; *Video 1*). In three reconstructions, two afferents projecting from different nerve trunks contacted Merkel cells within a single touch dome (*Figure 1E–E'*; *Video 2*). It is possible that these branches converged beyond the field of view. Alternatively, two distinct afferents might innervate Merkel cells in a minority of touch domes, as previously observed in rat (*Yasargil et al., 1988*; *Casserly et al., 1994*). We focused quantitative analysis on touch domes with single-afferent innervation (N = 15). Afferents displayed five to seven nested orders of branches. Arbor complexity, as represented by the highest branching order, did not correlate with Merkel-cell number (*Figure 1G*), which ranged almost fivefold (*Figure 1H*). Total branch number varied more than twofold between touch domes, and unmyelinated neurites accounted for most of this variation (*Figure 1H*). Quantities of myelinated branches and heminodes were more restricted and were independent of Merkel-cell counts (linear regression p=0.56 and 0.55, respectively). Most Merkel cells (>85%) were directly contacted by neurites, suggesting that they were incorporated into afferent arbors (*Figure 1I*). Similarly, ≥80% of terminal neurites were occupied by Merkel cells in most touch domes (*Figure 1I*). This quantitative analysis reveals a surprising degree of structural diversity in SAI-afferent end organs, particularly in the abundance of Merkel cell–neurite complexes. Given that the number of complexes exceeded heminodes within each arbor, we reasoned that individual heminodes must receive inputs from multiple Merkel cell–neurite complexes.

To determine how these complexes are arranged within an afferent's arbor, we analyzed the distribution of Merkel cell–neurite complexes among terminal neurites and heminodes. The number of terminal

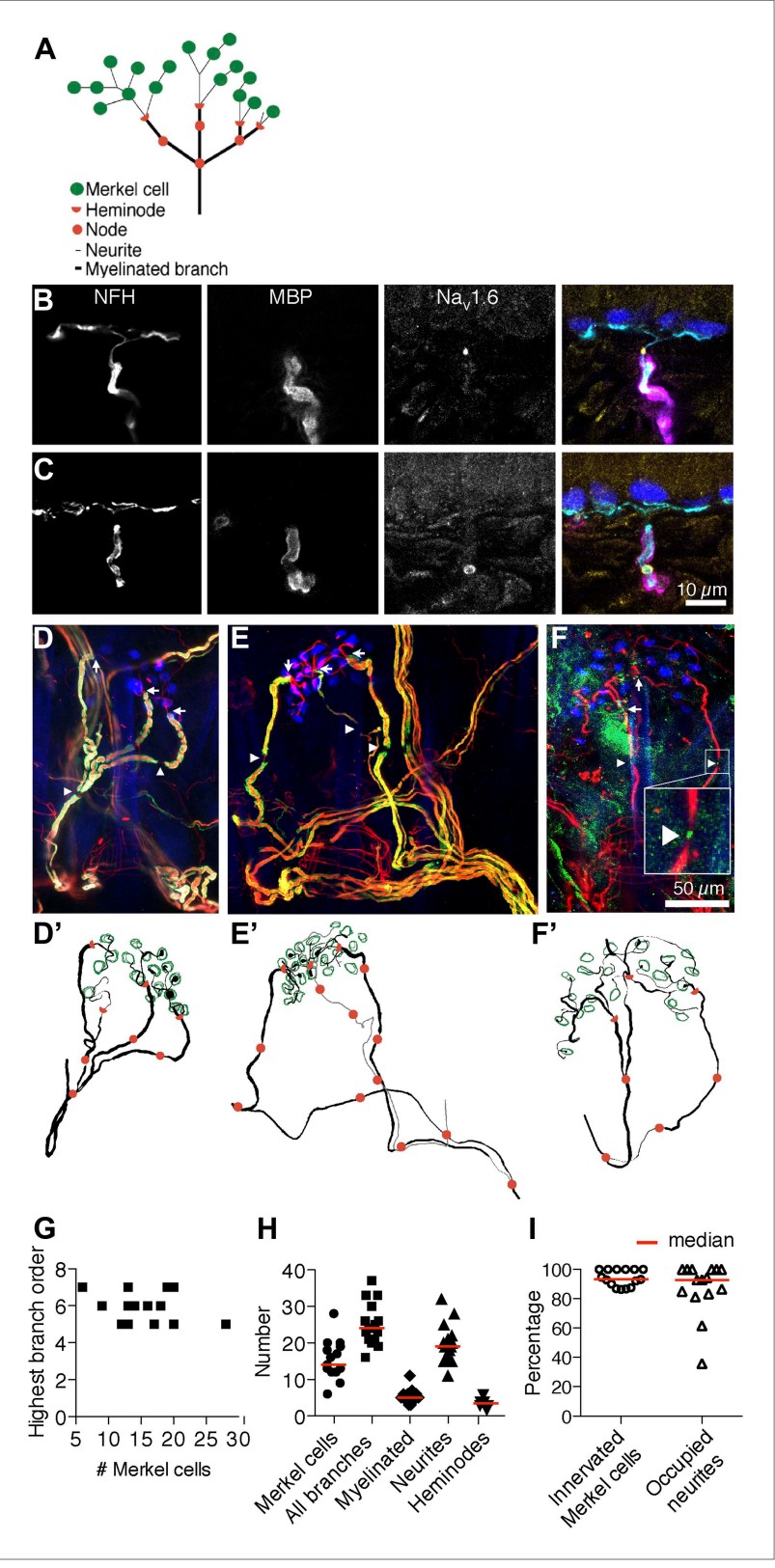

**Figure 1**. Morphometry of touch-dome afferents reveals diverse end-organ architectures. (**A**) Schematic of the SAI afferent's end organ. (**B** and **C**) SAI afferents, labeled with antibodies against Neurofilament-H (NFH; cyan) and Myelin Basic Protein (MBP; magenta), were identified by their connection to Keratin 8-positive Merkel cells

*Figure 1. Continued on next page*

*Figure 1. Continued*

(K8; blue) in touch-dome cryosections. The voltage-gated sodium channel Na$_V$1.6 (yellow) localized to heminodes (**B**) and nodes of Ranvier (**C**). Scale bar in **C** (10 µm) applies to **B**. (**D–F**) Projections of touch domes labeled in whole mount. (**D**) NFH (red), MBP (green) and K8 (blue) labeled Merkel cells contacted by a single myelinated afferent (see also *Video 1*) or (**E**) two afferent branches whose point of convergence was not identified (see also *Video 2*). Arrows: examples of heminodes; arrowheads: examples of nodes of Ranvier. (**F**) Na$_V$1.6 (green) identified heminodes and nodes in an NFH-positive afferent (red) innervating K8-positive Merkel cells (blue). Inset shows an expanded view of an Na$_V$1.6-positive node. Scale bar in **F** (50 µm) applies to **D–F'**. (**D'–F'**) Projections of 3D reconstructions of end organs shown above: afferent (black), Merkel cells (green), heminodes (red half-circles) and nodes (red circles). (**E'**) A non-converging branch is marked in gray. Note that this branch is thinner than other myelinated branches. (**G**) The highest branching order found in each SAI afferent arbor was independent of the number of Merkel cells contacted. (**H**) Morphometric quantification of reconstructed touch domes innervated by single afferents. (**I**) More than 80% of Merkel cells were contacted by neurites and a similar proportion of terminal neurites contacted Merkel cells (N = 15 touch domes from five mice in **G**–**I**). Red lines represent median values in **H** and **I**.

neurites emanating from each heminode was broadly distributed (*Figure 2A*). Most Merkel cells were arranged individually on terminal neurites (70%, N = 165, *Figure 2B,C*), although chains of three or more Merkel cells along individual neurites were occasionally observed (*Figure 2B,D*; *Ebara et al., 2008*). As with terminal neurites, the number of Merkel cell–neurite complexes per heminode was broadly distributed (*Figure 2E*). To quantify the degree of structural asymmetry within an arbor, heminodes were ordered by the size of their Merkel-cell clusters (*Figure 2F*). For each arbor, a plot of the number of complexes at each heminode was fitted with a linear regression, the slope of which captures the skewness of the cluster distribution (median = 2.3 complexes per heminode, interquartile range = 1.2–4.4; R$^2$ = 0.6–1.0). The degree of skew did not correlate with total number of Merkel cell–neurite complexes in the arbor (*Figure 2—figure supplement 1*). Together, these data indicate that spike initiation zones within each arbor integrate inputs from a variable number of mechanotransduction sites. We hypothesize that this asymmetric distribution accounts for features of the SAI afferent's physiological output.

## Electrophysiological recordings of SAI responses

Tactile afferents can differ in their touch-evoked response properties, including firing rate and mechanical sensitivity, which is the steepness of the stimulus–response relation. During sustained touch, SAI afferents produce a biphasic spike train characterized by high-frequency firing during stimulus onset (ramp phase) and low-frequency firing with a highly variable interspike interval (ISI) during sustained displacement (static phase). Touch-evoked firing rates and mechanical sensitivity vary considerably between individual SAI afferents (*Mountcastle et al., 1966*; *Goodwin et al., 1995*; *Wellnitz et al., 2010*).

To determine whether the number of Merkel cells in a receptive field can account for variability in firing properties, we measured SAI responses over a range of displacements (*Figure 3A,B*). By using a GFP-expressing Merkel-cell reporter strain (*Lumpkin et al., 2003*), we visualized the number of Merkel cells within each touch dome, which represents an upper bound on the number of Merkel cell–neurite complexes in the SAI-afferent arbor (N = 4 SAI afferents; *Figure 3B,C*). We first tested for a relationship between the total number of Merkel cells and the latency of first spikes, a measure that reliably conveys information about dynamic tactile stimuli (*Johansson and Birznieks, 2004*). For the first spike, which is

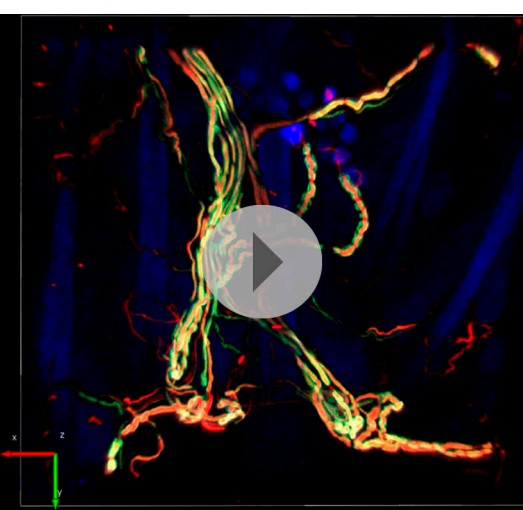

**Video 1**. Three-dimensional reconstruction and Neurolucida tracing of the touch dome in *Figure 1D*, which is innervated by a single SAI afferent.

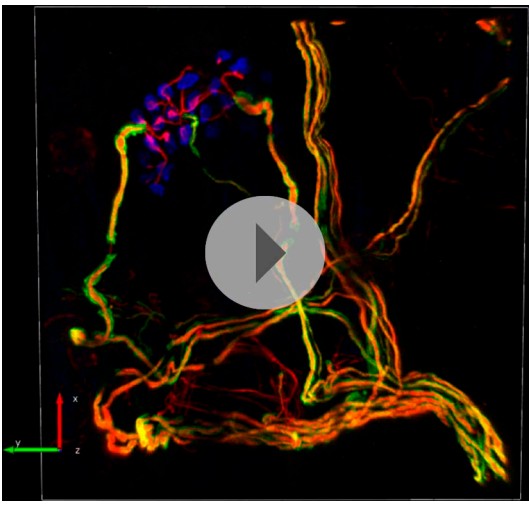

**Video 2**. Three-dimensional reconstruction and Neurolucida tracing of the touch dome in *Figure 1E*. This touch dome was innervated by three major branches, one of which did not converge within the imaging field. Note that this unbranched afferent is thinly myelinated and has a finer axonal diameter than typical SAI afferents.

independent of active zone resetting, the latency to reach firing threshold is expected to be inversely proportional to the number of transduction units activated by a given stimulus. Thus, touch domes with large Merkel-cell complements should have short first spike latencies compared with small touch domes. We grouped large touch domes (20 and 22 Merkel cells) and small touch domes (12 and 13 Merkel cells) to account for the possibility that up to 15% of Merkel cells were not innervated (*Figure 1I*). As predicted, first spike latencies were significantly shorter in large touch domes (mean ± SEM, 10.9 ± 1.6 ms, N = 57) compared with small touch domes (40.0 ± 14.5 ms, N = 60; p = 0.027; Student's *t* test, one-tailed) for suprathreshold stimuli. These data suggest that having more Merkel cell–neurite complexes in a touch dome results in a faster response during dynamic indentation. We also noted that the variance of first spike latencies was significantly higher in small touch domes (p<0.0001; two-sample F test, two-tailed), which suggests that SAI afferents with fewer transduction units display less reliable spike timing during dynamic stimulation.

We next analyzed displacement–response relations, which were fitted with single-exponential regressions. The time constant of the exponential fit, κ, was used to estimate an afferent's mechanical sensitivity and $Y_0$, the y-intercept, to estimate threshold firing rate (N = 4 afferents, *Figure 3B*). This analysis confirmed that mechanical sensitivity differed significantly between SAI afferents innervating mouse touch domes (κ = 2.1–14.2 $mm^{-1}$; p < 0.0001, Extra sum-of-squares F test); however, this measure did not scale with total Merkel-cell number.

SAI afferents have been reported to innervate more than one touch dome in cats and neonatal mice (*Tapper, 1965*; *Iggo and Muir, 1969*; *Woodbury and Koerber, 2007*). As our 3-mm probe tip is large enough to cover several touch domes, it is possible that mechanical sensitivity scales with the number of touch domes innervated by an individual afferent. To rule out this possibility, we manually probed the skin's surface to identify all receptive fields for each SAI afferent. For computational modeling, we analyzed SAI afferents whose receptive fields were limited to single touch domes (*Figure 3*). To determine whether single touch-dome innervation is typical of SAI afferents in adult mice, we analyzed a larger dataset of SAI afferent recordings that was not biased for receptive field structure (N = 27 afferents). We found that 19 SAI afferents innervated individual touch domes, six innervated two touch domes each and two afferents innervated three touch domes each. Thus, the percentage of SAI afferents that innervate multiple touch domes in the hindlimb of adult mice (30%) is much lower than that reported in cats (>60%; *Tapper, 1965*; *Iggo and Muir, 1969*) or mouse neonatal back skin (3/4 SAI afferents; *Woodbury and Koerber, 2007*).

We considered two additional factors that might contribute to the observed differences in SAI-afferent sensitivity. First, skin mechanics did not account for these differences because displacement–force relations were indistinguishable between these recordings (*Figure 3D*). A second possibility is that a touch dome might be innervated by multiple SAI afferents. In that case, the number of Merkel cells contacted by each afferent would be lower, resulting in reduced firing rates and mechanical sensitivities. This scenario is likely to apply to only a minority of mouse touch domes because >80% of reconstructed touch domes were innervated by a single myelinated afferent (*Figure 1*); therefore, we sought to identify additional structural features that might account, in part, for differences in touch-evoked firing. We focused on the grouping of Merkel cell–neurite complexes to heminodes because this feature varied substantially between SAI afferents.

## Computational modeling of touch-receptor end organs

We used predictive computational modeling to test functional consequences of the asymmetric distribution of mechanotransduction sites in SAI afferents. This approach affords the ability to analyze

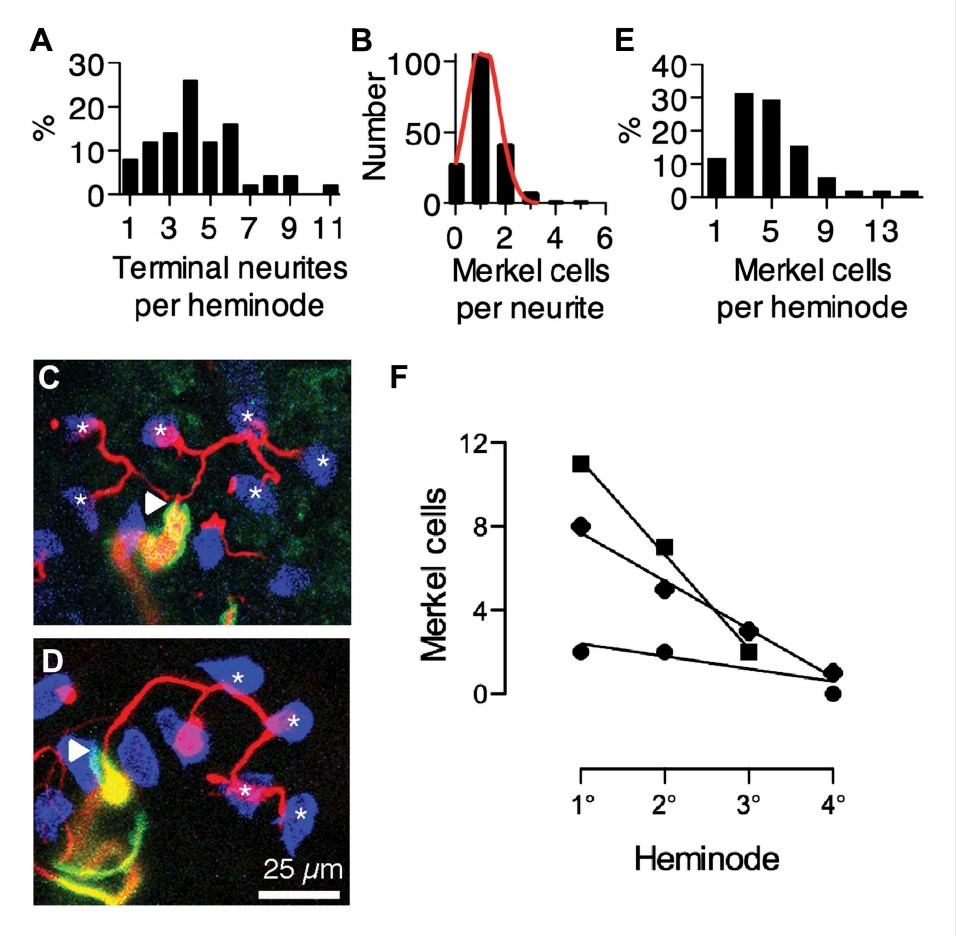

**Figure 2**. Merkel cell–neurite complexes are asymmetrically distributed between heminodes. (**A**) Distribution of terminal neurites per heminode (N = 219 neurites). (**B**) Histogram of the number of Merkel cells contacted by each terminal neurite (N = 226 Merkel cells). Red: Gaussian fit ($R^2$ = 0.99). (**C**) Confocal projection of six terminal neurites contacting individual Merkel cells (asterisks). (**D**) A projection of a single terminal neurite contacting a chain of four Merkel cells (asterisks). Arrowheads denote heminodes and scale bar (25 μm) applies to **C** and **D**. (**E** and **F**) The distribution of Merkel cell–neurite complexes per heminode from pooled touch-dome afferents (**E**; N = 51 heminodes from 15 touch domes) and within individual tactile arbors (**F**). In **F**, number of Merkel cells at each heminode from three touch domes is plotted from the largest, or primary (1°), cluster to the smallest, quaternary (4°), clusters. Representative touch domes across the skew range are shown and linear regressions are plotted (slopes = −0.6, −2.3 and −4.5, $R^2$ = 0.6, 0.99, 1.0). See also *Figure 2—figure supplement 1*.
The following figure supplements are available for figure 2:

**Figure supplement 1**. Skew values for each touch dome plotted vs the number of Merkel cell-neurite complexes in the arbor.

the effects of neuronal architecture on predicted firing patterns by systematically manipulating potential end-organ configurations. Our models assume that each Merkel cell–neurite complex serves as a mechanotransduction unit capable of producing receptor currents and that resulting signals sum to initiate spikes at heminodes.

To represent the SAI afferent's end organ in the skin, we constructed a novel network model comprising three modules, as detailed in 'Materials and Methods' (***Figure 4A,B***). First, a finite element model (FEM) of skin mechanics transformed skin displacement into strain energy density (SED) at the location of mechanotransduction units. Second, a sensory transduction module transformed SED values into receptor currents. To account for the biphasic SAI response, the transduction function contained a dynamic component proportional to the rate of change in SED and a static component

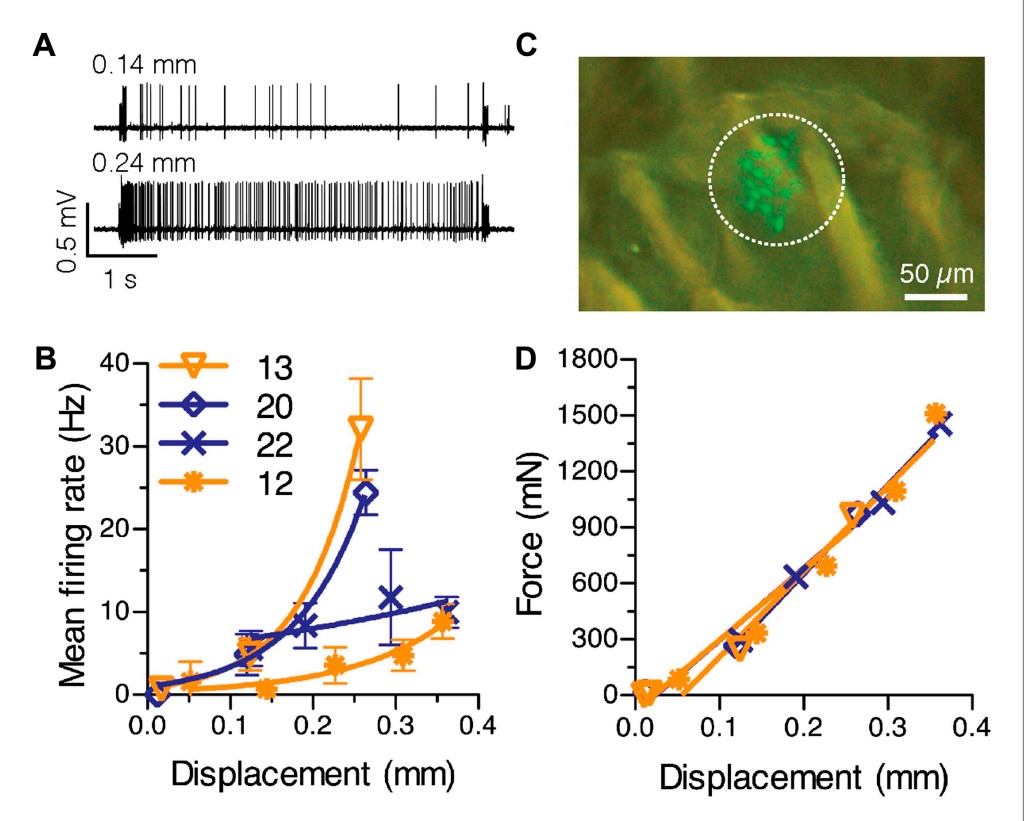

**Figure 3**. Physiological response properties vary between mouse SAI afferents. (**A**) Extracellular recordings from an SAI afferent stimulated at two displacement magnitudes demonstrates the biphasic SAI response, which is characterized by high-frequency firing during the ramp phase, as well as slow adaptation and variable spike timing during the static phase. (**B**) Displacement–response relations from individual SAI afferents. Legend indicates the number of Merkel cells in each touch dome quantified based on GFP fluorescence. Responses from receptive fields with large end organs (blue) and small end organs (orange) are shown. Firing rates during the static phase are plotted (mean ± SD, N = 3–12 stimuli per displacement magnitude). Data were fitted with single exponentials to estimate mechanical sensitivity ($\kappa$) and threshold firing rate ($Y_0$; $R^2$ = 0.63–0.99). (**C**) Merkel cells (green) from *Atoh1/nGFP* transgenic mice selectively express GFP. The receptive field of the SAI afferent in **A** is shown (dotted line). (**D**) Force-displacement relations measured during the recordings shown in **B**. Skin mechanics were indistinguishable between these recordings.

proportional to SED. A noise term accounted for the SAI afferent's characteristic ISI variablility during the static phase of stimulation (*Figure 4C*). The transduction function predicted an adapting receptor current, *I(t)*, whose form is consistent with those recorded from a wide range of mechanosensory receptor cells, including inner-ear hair cells (*Eatock et al., 1987*), *Drosphila* bristle neurons (*Walker et al., 2000*), and somatosensory neurons in vitro (*Lechner et al., 2009*). Third, a neural dynamics module, consisting of an array of leaky integrate-and-fire (LIF) models that represent spike initiation zones, summed receptor currents and, at threshold, produced spike times. A unique feature of this network model is that it allows for reconfigurable transduction functions that represent asymmetrically grouped Merkel cell–neurite complexes at spike initiation zones.

We first created a model of a reconstructed SAI arbor (*Figure 1D'*) containing four heminodes with clusters of eight, five, three and one Merkel cell–neurite complexes. The resulting model had four spike initiation zones with transduction-unit groupings of {8, 5, 3, 1} (*Figures 1A and 4A*). The model's spike-timing predictions were fitted to a prototypical mouse SAI response (*Figure 4C,D*). To derive the prototypical SAI response, we performed regressions of ramp- and static-phase responses from four mouse SAI afferents analyzed in aggregate (*Figure 4—figure supplement 1*). The model produced spike times that reproduced the dynamics of SAI responses over a range of stimulus conditions, including different displacement magnitudes and ramp accelerations. Firing properties that were

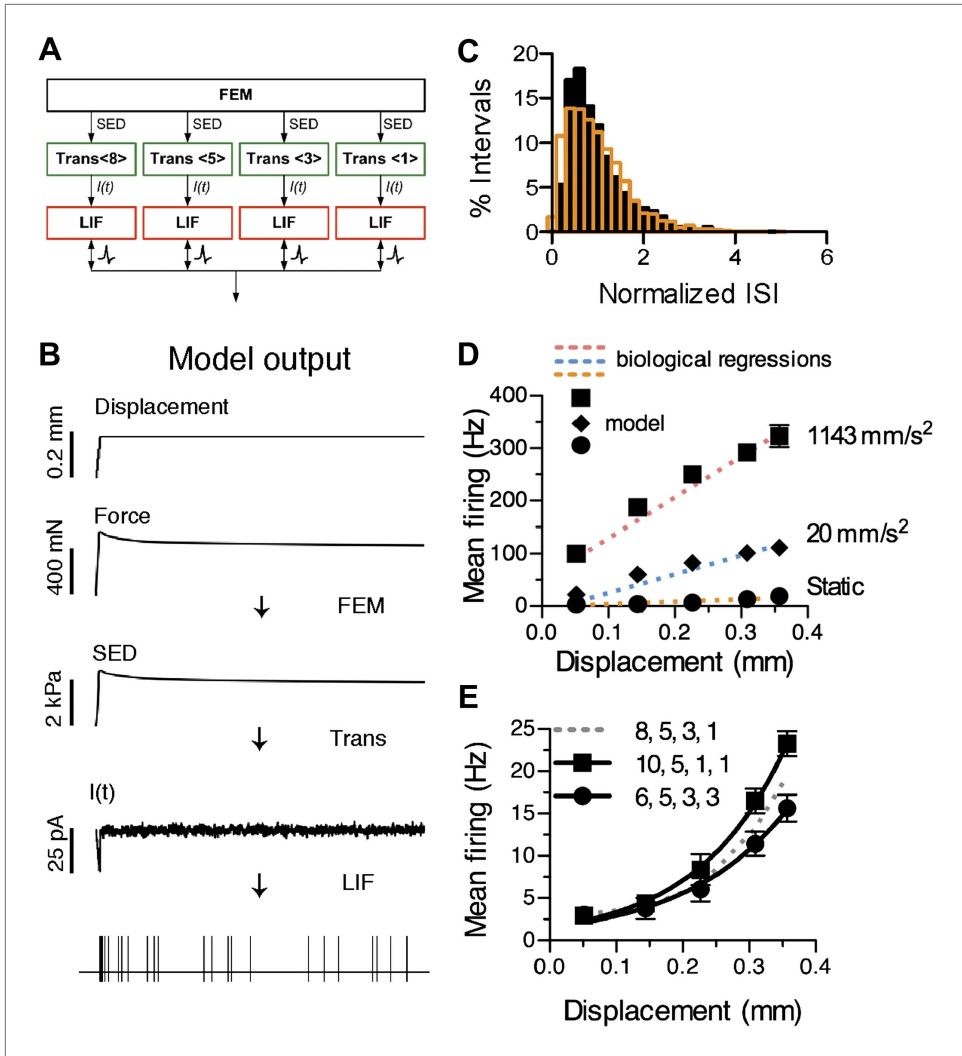

**Figure 4**. Computational modeling recapitulates characteristic features of the SAI response. (**A**) The network model configuration for the reconstructed SAI afferent in *Figure 1D'*. (**B**) Data flow through computational models and example outputs from each module: a finite element model (FEM) produces strain energy density (SED) at transduction units, transduction functions (Trans <# merkel cell–neurite complexes>) predict transduction currents ($I(t)$) and a leaky integrate-and-fire (LIF) array produces spike times. (**C**) The model's predicted spike-timing variability, assessed by the distribution of normalized ISIs during static-phase responses (black bars: N = 1,591 intervals), corresponded to the skewed Gaussian distribution previously reported for mouse SAI afferents (orange bars: N = 3,348 intervals from 11 afferents; *Wellnitz et al., 2010*). To compare ISIs across a range of displacement magnitudes, each ISI was normalized to the mean interval for its stimulus. (**D**) Simulated firing rates (black symbols) from the model configuration in **A** were fitted to linear regressions of ramp-phase (blue dotted line: ramp acceleration = 20 mm·s$^{-2}$, pink dotted line: ramp acceleration = 1143 mm·s$^{-2}$) and static (orange dotted line) responses pooled from the SAI afferents shown in *Figure 3B*. Goodness of fit = 0.96 (fractional sum of squares). (**E**) Displacement–response relations from models configured with different primary cluster sizes. All configurations had 17 total transduction units and four spike initiation zones. Mean firing rates during the static phase of displacement are plotted (mean ± SD, N = 15 simulations per displacement). Displacement-response curves were compared by fitting with exponential regressions (R$^2$ ≥ 0.99). Increasing or decreasing primary cluster size by two transduction units significantly changed the best fits (8 vs 10: p=0.004; 6 vs 8: p=0.017, extra sum-of-squares F test). Legend indicates the distribution of transduction units at spike initiation zones.

The following figure supplements are available for figure 4:

**Figure supplement 1**. Linear regression analysis of pooled responses from the four SAI afferents in *Figure 3B* (denoted by symbols; N = 3–12 stimuli per displacement magnitude).

well-fitted by the model included a high-frequency response at displacement onset, an adapted firing rate during the static phase and higher firing rates with increasing displacement magnitude and acceleration (*Figure 4D*). Static- and ramp-phase response profiles were fitted with an $R^2$ of 0.96, as measured by fractional sum of squares (*Figure 4D*).

## Computational simulations predict that arbor architecture can account for differences in mechanical coding between SAI afferents

Taking advantage of the reconfigurable computational model, we tested the hypothesis that the relative distribution of transduction units at spike initiation zones influences touch-evoked firing. We compared predicted firing rates during sustained displacement for different end-organ configurations with 17 Merkel cells (*Figure 4E*). From the initial configuration of {8, 5, 3, 1}, we found that moving only two transduction units, to yield groupings of {10, 5, 1, 1} or {6, 5, 3, 3}, were sufficient to significantly alter the shape of simulated displacement–response relations. Increasing the primary cluster to 10 units increased firing rates for suprathreshold stimuli by 20%. Conversely, moving two transduction units from the largest group to the smallest decreased predicted firing rates for supra-threshold displacements by 25%.

Similar results were observed for three additional modeled arbors configured to represent the range of anatomical features observed in reconstructions (*Table 1*). These models differed in their numbers of spike initiation zones (3–5) and transduction units (13–20). Simulated firing rates were also enhanced, though to a lesser extent, when largest clusters were held constant and secondary clusters were increased (*Table 1*). On average, firing rates increased 7.2% per transduction unit added to a primary cluster and 2.8% per transduction unit added to a secondary cluster. Collectively, these models predict that touch-evoked firing is increased when Merkel cell–neurite complexes are arranged in a skewed distribution among heminodes.

Although these studies predict that the arrangement of transduction units can in part set the coding properties of tactile afferents, we reasoned that the number of transduction units must also impact firing rate, since activating additional units will more readily bring the membrane potential to spike threshold. The interaction of these two parameters was examined by systematically adding transduction units to four prototypical models to increase end-organ size. For each arbor, two strategies were used to 'fill up' clusters until they equaled the size of the primary cluster. In a first set of simulations, transduction units were progressively added to secondary clusters. Alternatively, transduction units were added to smallest clusters (*Figure 5A*). The first strategy, which skewed the distribution of transduction units, boosted firing rates more than equalizing the distribution with the second strategy (*Figure 5A*). For example, increasing transduction units from 17 to 24 augmented responses on average by 39% when they were added to secondary clusters but only by 21% when they were more evenly distributed (*Figure 5A*, Arbor 1). This effect was consistently observed across prototypical models (*Figure 5A*, Arbors 2–4). Thus, our simulations predict both the number of transduction units and their arrangement within the arbor regulate SAI afferent firing properties.

Finally, we asked whether a small arbor with few transduction units can display a heightened mechanical sensitivity compared with a large arbor, as we observed in electrophysiological recordings

**Table 1.** Effects of primary and secondary cluster size on firing rate

| Model arbor # | Merkel-cell number | Grouping 1 | Grouping 2 | ΔPrimary group | ΔSecondary group | % Firing Rate Δ |
|---|---|---|---|---|---|---|
| 1 | 17 | {**6**, 5, 3, 3} | {**10**, 5, 1, 1} | 4 | – | 39 |
| 1 | 17 | {8, **3**, 3, 3} | {8, **7**, 1, 1} | – | 4 | 15 |
| 2 | 20 | {**6**, 6, 4, 2, 2} | {**9**, 6, 3, 1, 1} | 3 | – | 18 |
| 2 | 20 | {7, **4**, 4, 3, 2} | {7, **7**, 4, 1, 1} | – | 3 | 9 |
| 3 | 13 | {**4**, 4, 3, 2} | {**6**, 4, 2, 1} | 2 | – | 12 |
| 3 | 13 | {5, **3**, 3, 2} | {5, **5**, 2, 1} | – | 2 | 4 |
| 4 | 13 | {**5**, 4, 4} | {**7**, 4, 2} | 2 | – | 14 |
| 4 | 13 | {6, **4**, 3} | {6, **6**, 1} | – | 2 | 5 |

Bold values indicate the group whose number was changed in the computational experiment.

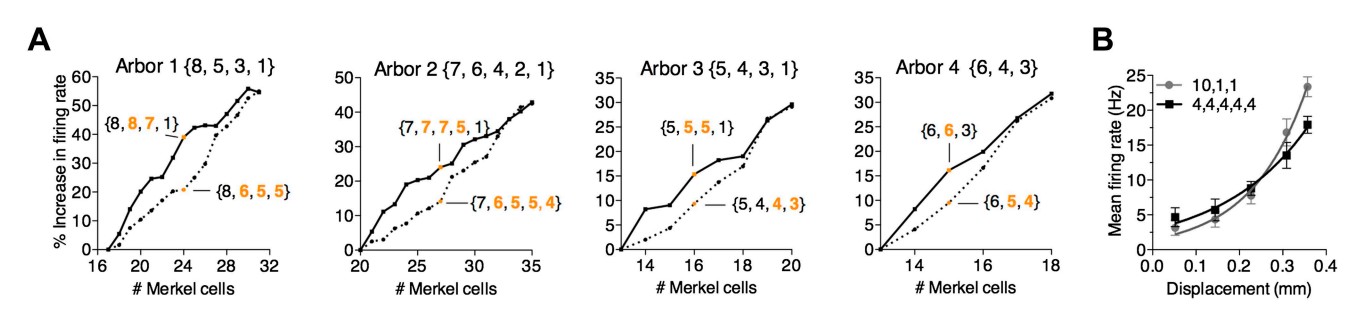

**Figure 5**. A survey of computational parameter space predicts that the number and arrangement of mechanosensory transduction units modulates SAI-afferent firing properties. (**A**) Two strategies for adding transduction units to an SAI-afferent arbor were tested in four independent model end organs (Arbors 1–4). Arbor configurations differed in number of spike initiation zones (3–5) and initial end-organ sizes (13–20). Transduction units were added progressively to either secondary (solid lines) or smallest clusters (dashed lines). Orange symbols highlight examples from the two strategies after adding multiple transduction units. Example cluster arrangements are indicated in brackets. Clusters changed from the initial configuration are indicated in orange font. The percent change in firing rate from baseline configuration is plotted. (**B**) Comparison of displacement–response relations (mean ± SD, N = 15 stimuli per displacement magnitude) for two model configurations indicated in brackets: a skewed distribution of 12 transduction units among three spike initiation zones (gray) and an equal distribution of 20 transduction units among five spike initiation zones (black). Simulation results were fitted with single exponential equations ($R^2 \geq 0.99$). The mechanical sensitivity of the small end organ was predicted to be significantly greater than that of the large end organ ($\kappa$ = 7.7 and 5.0, respectively, p=0.005, extra sum-of-squares F test).

(*Figure 3B*). Two arbor configurations were computationally compared (*Figure 5B*). The first had 12 transduction units asymmetrically grouped at three spike initiation zones (skew = 4.5, {10,1,1}). The second had 20 transduction units evenly distributed among five spike initiation zones (skew = 0; {4,4,4,4,4}). Despite having 40% fewer transduction units, the skewed grouping strategy produced a significantly higher mechanical sensitivity than the evenly distributed end organ ($\kappa$ = 7.7 and 5.0, respectively, p = 0.005; *Figure 5B*). This was not simply due to increased firing rates for suprathreshold stimuli, as firing rates at threshold were significantly lower for the small end organ compared with the large one ($Y_0$ = 1.48 and 2.92 Hz, respectively; p = 0.016). Thus, these simulations demonstrate that a tactile afferent with few transduction units can achieve high touch sensitivity by unevenly grouping transduction units to action potential initiation sites.

## Discussion

In nervous systems ranging from *C. elegans* to mammals, touch receptors display a rich array of specialized end organs that correlate with distinct physiological functions; however, little is known about how specific architectural features govern neuronal firing patterns (*Chalfie, 2009*). In this study, we combined neuroanatomy, electrophysiology and computational modeling to identify structural features of a mammalian touch receptor that have the potential to impact neuronal firing. Our morphometric analysis extends previous studies that visualized arbors of neonatal SAI afferents by tracer iontophoresis (*Woodbury and Koerber, 2007*) and surveyed cutaneous afferents in mouse hairy skin by sparse genetic labeling (*Li et al., 2011*; *Wu et al., 2012*). To our knowledge, this study is the first to construct computational models of mammalian tactile afferents that are constrained with structural parameters quantified from intact tactile end organs. Computational results predict that asymmetric clustering of transduction sites at spike initiation zones regulates mechanosensory coding in a branched tactile afferent. These results generate testable hypotheses and highlight the integral role of peripheral neuronal structure in somatosensory signaling.

### Structural and molecular analysis of SAI afferents

Morphometric analysis of SAI afferents revealed multiple heminodes, the anatomical correlates of spike initiation zones in myelinated tactile afferents (*Loewenstein and Rathkamp, 1958*). Our results are the first to localize Na$_V$1.6 in tactile end organs, which extends previous reports that identified this isoform in unmyelinated nociceptors and as the principal sodium channel at central and peripheral nodes (*Caldwell et al., 2000*; *Black et al., 2002*). The observation that Na$_V$1.6 localizes to almost all heminodes and nodes within an end organ suggests that this ion channel plays an important role in

spike initiation and integration in tactile afferents; however, other $Na_V$ isoforms might also be found at these sensory endings. SAI afferents display unusually high instantaneous firing frequencies exceeding 1000 Hz (*Iggo and Muir, 1969*). Thus, it is notable that $Na_V1.6$ confers rapid sodium channel kinetics and mediates resurgent currents, which facilitate high-frequency firing, in DRG neurons (*Herzog et al., 2003*; *Cummins et al., 2005*).

Our results also reveal how epidermal Merkel cells are incorporated into tactile arbors. Our quantification of Merkel cells in adult touch domes is consistent with that reported at E18.5 (*Lesko et al., 2013*). Although most terminal neurites innervate single Merkel cells, the Gaussian distribution of these connections suggests that they are formed through probabilistic rather than deterministic mechanisms. Interestingly, touch-dome afferents display exuberant terminal branching in *Atoh1* knockout mice, which lack Merkel cells (*Maricich et al., 2009*). These findings suggest that targeting and maintenance of touch-dome innervation is independent of Merkel cell-derived signals. Instead, we propose that Merkel-cell contacts are required for appropriate sprouting and/or pruning of touch-dome arbors. Additional studies are needed to determine whether neurites induce Merkel-cell differentation or whether Merkel cell-derived signals establish stable neuronal connections.

## Computational modeling of intact tactile end organs

Although neural dynamics and skin mechanics are tied together in vivo, tactile afferent neural dynamics and skin mechanics models have largely been used in isolation. Thus, the models presented here are the first to computationally represent the sequence of key events in tactile encoding: the conversion of touch at the skin's surface to mechanical distortion at tactile receptors, mechanoelectrical transduction and spike initiation at heminodes. This representation was achieved by combining three model sub-components, each of which extends previous efforts to model tactile responses.

Previous studies have taken one of three general modeling approaches. First, empirical models such as those of Goodwin and Wheat use simple regression functions to abstract away the roles of skin and mechanoreceptors. These models focus on the role of noise and receptor co-variance in predicting population responses that align with psychophysical studies (*Goodwin and Wheat, 1999*, *2002*). Second, skin mechanics models use finite elements and continuum mechanics to represent how surface forces propagate to tactile end organs, but abstract away neural dynamics by using scaling functions to predict firing rates. A limitation is that these models only predict firing rates for steady-state stimuli as opposed to spike times. Finally, neural dynamics models convert receptor currents to spike timing but disregard the skin's role in shaping end-organ output. For vibratory stimuli delivered with a skin-attached probe, this simplification is reasonable as the skin's role is minimal when it follows probe movement closely. By contrast, viscoelastic skin relaxation occurs during sustained touch stimuli, such as those encoded by SAI afferents (*Cohen et al., 1999*).

In this study, we modeled skin mechanics using hyper- and visco-elastic material models with parameters fitted to values from mammalian tissues. Material models were validated against force-displacement data measured during ex vivo skin-nerve recordings and extend a previous study that used a linear elastic model (*Lesniak and Gerling, 2009*). Although parameter values were chosen within reasonable ranges for mouse skin, future models could be refined by employing recent compressive measurements of mouse skin (*Wang et al., 2013*). It is also possible that the material properties of the touch dome itself differ from surrounding epidermis.

In addition to combining skin and neuron models, the network model presented here extends previous neural dynamics models. Prior models have employed rate-sensitive transduction functions and LIF functions to make spike timing predictions by calculating SAI membrane potential as a function of vibration frequency and magnitude (*Freeman and Johnson, 1982a*; *1982b*; *Kim et al., 2009*; *Kim et al., 2010*). Previous neural dynamics models have not accounted for end-organ size, neuronal branching or multiple sites of spike initiation. In this study, we utilized multiple, resettable LIF models in conjuction with transduction functions parameterized by the number of Merkel cell–neurite complexes. We introduced noise at the level of current within the transduction functions. Another approach to recreate the SAI afferent's irregular interspike intervals could be to introduce probabilistic firing and adaptive thresholds at spike initiation zones, similar to that done for vibratory stimuli (*Jahangiri and Gerling, 2011*; *Dong et al., 2013*). By employing an array of spike initiation zones, our model allows zone resetting upon action potential firing, consistent with that observed for SAI afferents and other myelinated somatosensory afferents (*Adrian and Zotterman, 1926a*; *Horch et al., 1974*).

## End-organ asymmetry and the driver effect

Although Merkel cells make one-to-one connections with neurites, we found that these complexes were asymmetrically distributed between heminodes. Primary clusters of Merkel cell–neurite complexes, converging on a single heminode, sometimes contained ≥50% of Merkel cells in the entire arbor. In computational simulations, changing the primary cluster size by as few as two Merkel cell–neurite complexes significantly altered afferent firing. In this study, anatomical reconstructions were performed on freshly excised tissue to ensure that tissue morphology was well preserved for quantitative morphometry. Thus, anatomical reconstructions and electrophysiological recordings were achieved with different SAI afferents. Future studies to record and reconstruct individual sensory afferents are needed to directly test the model's predictions. Nonetheless, our findings provide an anatomical correlate and a plausible biological mechanism to explain the driver effect observed in branched sensory afferents (*Lindblom and Tapper, 1966*; *Horch et al., 1974*; *Fukami, 1980*).

Theoretically, the most sensitive receptor configuration consists of a single cluster of transduction sites connected to a single heminode, as found in invertebrate stretch receptors (*Edwards and Ottoson, 1958*). What is the biological advantage of distributing transduction complexes among multiple heminodes, as we observed for SAI afferents? First, given that the skin is our body's protective covering, this arrangement could serve as a safety feature by increasing robustness to injury. Second, for cutaneous afferents with large receptive fields, multiple spike initiation zones ensure high-fidelity signal propagation from branches located millimeters apart (*Li et al., 2011*; *Wu et al., 2012*). For example, an SAI afferent can innervate 2–5 touch domes, each spaced ~0.7 mm apart (*Tapper, 1965*; *Iggo and Muir, 1969*; *Woodbury and Koerber, 2007*; *Wellnitz et al., 2010*; *Li et al., 2011*). We also noted that heminodes were located within 19–41 μm of lanceolate endings innervating hair follicles (N = 7 end organs). As individual rapidly adapting afferents can innervate tens to hundreds of hair follicles (*Li et al., 2011*; *Wu et al., 2012*), this observation suggests that spike initiation zones in close proximity to end organs is a general feature of myelinated tactile afferents. Third, distinct clusters might extend the receptor's range of sensory coding (*Eagles and Purple, 1974*). For example, individual muscle-spindle afferents are proposed to encode both dynamic and static stimuli by innervating distinct structures called bag and chain fibers (*Quick et al., 1980*; *Banks et al., 1997*). Although our models assume equivalent transduction units, it is possible that populations of Merkel cell–neurite complexes are likewise tuned to different stimulus features.

Our reconstructions suggest that some touch domes might be innervated by multiple sensory afferents, and this likely contributes to the wide range of firing properties observed for SAI afferents. SAI afferents with overlapping receptive fields have been described in rat touch domes located at dermatome borders (*Yasargil et al., 1988*; *Casserly et al., 1994*). Moreover, human touch domes are proposed to be innervated by distinct types of sensory afferents (*Reinisch and Tschachler, 2005*). In two touch domes, we observed both a typical SAI afferent and a thinly myelinated, unbranched afferent that contacted Merkel cells. We speculate that these are Aδ afferents based on their thin myelin sheaths and axonal diameters. The development of selective molecular markers is needed to understand how signals from distinct touch-dome neurons are integrated in the central nervous system.

In monkey and human fingerpads, SAI afferents have non-uniform receptive fields with multiple hot spots that display higher firing rates than surrounding areas (*Phillips et al., 1992*; *Vallbo et al., 1995*). These hot spots can explain why the resolution of primate SAI afferents is smaller than their receptive field sizes (*Phillips et al., 1992*). Johnson and colleagues hypothesized that hot spots correlate with the locations of individual Merkel cell–neurite complexes; however, their observations of 3–5 hot spots per receptive field in primates (*Phillips et al., 1992*) coincides well with our finding of 2–6 heminodes per touch-dome SAI afferent. Thus, we propose that the structural basis of a receptive-field hot spot is a cluster of Merkel cell–neurite complexes at a heminode. As SAI-afferent receptive field sizes and skin structure differ markedly between primate plantar skin and mouse touch domes, these observations suggest an organizing principle for SAI-afferent end organs across species and skin sites. Confirming the anatomical basis of hot spots will require the development of new transgenic mice to visualize individual SAI afferent branches during intact electrophysiological recordings, as well as microstimulation techniques to deliver controlled punctate stimuli to individual Merkel cells or Merkel cells clustered at single heminodes. To model such punctate stimuli will require building, validating and experimentally constraining new finite element models with a finer discretized mesh than the one used here.

## Potential consequences of fiber-to-fiber variability among SAI afferents

Individual SAI afferents are capable of representing shapes, edges and curvature with high fidelity (*Johnson and Lamb, 1981*; *Phillips and Johnson, 1981*; *Johnson and Hsiao, 1992*). Nonetheless, tactile qualities are conveyed to the central nervous system by an array of afferents distributed across the skin. To faithfully encode spatial features at the population level, one would expect SAI afferents to display uniform firing properties. Instead, touch-evoked firing rates, first spike latencies and mechanical sensitivity varied widely between SAI afferents in mouse touch domes, which corroborates previous reports of SAI afferent-to-afferent variability in monkey and human fingerpads (*Phillips and Johnson, 1981*; *Goodwin et al., 1995*, *1997*; *Goodwin and Wheat, 1999*). Thus, this variability is likely to be a general feature of mammalian SAI afferents.

How might the central nervous system cope with this large variation in firing properties among a single class of tactile afferents? Simulated population responses predict that such differences will distort the representation of an object's spatial features (*Goodwin and Wheat, 1999*). It is possible that the central nervous system introduces a scaling factor to compensate for peripheral distortion (*Goodwin et al., 1995*; *Goodwin and Wheat, 1999*). Alternatively, the nervous system could take advantage of this variability to efficiently transfer information. For example, having a variety of SAI-afferent sensitivities might extend the dynamic range of the SAI-population response to sustained pressure. Moreover, since some SAI afferents innervate two or more touch domes, it is possible that variations in end-organ structure confer different firing properties to individual touch domes (*Lindblom and Tapper, 1966*). A moving stimulus will sequentially activate such receptive fields. In that case, one could envisage that distinct firing patterns arising from these receptive fields could provide a mechanism for tracking movement at the single-afferent level.

We propose that variability in SAI end-organ structure observed in this study is the outcome of homeostatic mechanisms engaged during normal skin remodeling. Merkel cells renew within touch domes and whisker follicles (*Van Keymeulen et al., 2009*; *Woo et al., 2010*; *Doucet et al., 2013*). Moreover, hair-growth cycles are accompanied by innervation changes (*Peters et al., 2001*; *Shimomura and Christiano, 2010*). We speculate that SAI-afferent arbors with their Merkel-cell complements are likewise dynamic throughout adulthood. Our simulations predict that altering the number or arrangement of Merkel cells changes touch-evoked firing. Thus, our findings raise the possibility that the nervous system employs homeostatic mechanisms to achieve reliable signaling from individual touch receptors. This work sets the stage to identify molecular mechanisms that cutaneous afferents use to maintain signaling fidelity during normal tissue remodeling and in the context of repair.

## Materials and methods

### Animals

Animal use was conducted according to guidelines from the National Institutes of Health's *Guide for the Care and Use of Laboratory Animals* and was approved by the Institutional Animal Care and Use Committees of Baylor College of Medicine and Columbia University Medical Center.

### Immunostaining and microscopy

Skin was depilated (Surgi-cream; Ardell, Los Angeles, CA) and dissected from the proximal hind limb of female *Atoh1/nGFP* transgenic mice (8–10 weeks of age). This location was chosen to match the site of electrophysiological recordings in ex vivo skin-saphenous nerve preparations. Tissue was fixed in 4% paraformaldehyde (PFA) or, for staining with Na$_V$1.6 antibodies, in 2% PFA in a sodium-acetate buffer (pH 6). For section staining, the skin was cryopreserved in 30% sucrose, frozen and cryosectioned at 25 μm. The sections were incubated overnight at room temperature in primary antibodies: rat anti-K8 (TROMA-I; Developmental Studies Hybridoma Bank, Iowa City, Iowa), chicken anti-NFH (AB5539; Millipore, Billerica, MA), rabbit anti-MBP (ab40390; Abcam, Cambridge, MA) and rabbit anti-Na$_V$1.6 (from MNR). The specificity of Na$_V$1.6 antibodies was previously validated as described (*Rasband et al., 2003*) and control experiments lacking primary antibody demonstrated the specificity of immunoreactive puncta at nodes. Secondary goat Alexa Fluor-conjugated antibodies (Invitrogen, Carlsbad, CA) directed against rat (Alexa Fluor 594; A11007), chicken (Alexa Fluor 647; A21449) or rabbit (Alexa Fluor 488; A11008) IgG were incubated for 1 h at room temperature. Whole-mount immunostaining was performed as reported (*Li et al., 2011*) with antibodies listed above. Tissue was incubated at room temperature with primary antibodies for 72–96 h and secondary antibodies for 48 h. The tissue was imaged on a Zeiss Exciter confocal microscope with 20X, 0.8 NA or 40X, 1.3 NA objective lenses.

### 3D reconstructions

Confocal image stacks were imported into Neurolucida (MBF Bioscience, Williston, VT) and traced in three dimensions. Images were prepared for publication in ImageJ (*Schneider et al., 2012*) or Photoshop (Adobe, Mountain View, CA). Two independent observers quantified heminodes, nodes, branches and Merkel cell-neurite complexes by stepping though optical sections in each reconstruction.

### Electrophysiology

Single-unit SAI afferent recordings from mouse ex vivo skin-saphenous nerve preparations were performed as previously described (*Wellnitz et al., 2010*). Recordings were made from adult *Atoh1/nGFP* transgenic mice to visualize Merkel cells within the intact skin. Mechanical stimuli were delivered via a ceramic cylindrical probe (3-mm tip diameter) mounted on a displacement-controlled indenter. Stimuli were 5-s displacements ranging from 0.01 to 0.36 mm and applied in a randomized order. The skin's reactive force was monitored with a load cell mounted on the indenter. Ramp-phase firing rates were calculated by dividing the number of spikes during the ramp phase by the ramp duration (i.e., the time period from probe contact with skin to final displacement). This calculation differs from a previous study that analyzed dynamic firing during the first 200 ms of stimulation, including the ramp phase and the period of rapid adaptation (*Wellnitz et al., 2010*). Static firing rate was defined as the number spikes per second calculated during a 2.5-s window after the stimulus probe had reached its commanded depth. This time window excludes the period of rapid adaption that follows the dynamic phase of the SAI response.

### Computational modeling

#### Skin mechanics

The skin was represented with a hyperelastic (Mooney-Rivlin) and viscoelastic (Prony Series) finite element model. A two-dimensional axisymmetric mesh represented the epidermis (17-µm thick), dermis (224-µm thick), subcutaneous tissue (101-µm thick) and the elastic substrate under the skin in the electrophysiology recording chamber to accurately represent experimental conditions. ABAQUS Standard (ver. 6.6) was used to create the model's geometry and mesh and was used for the FE analysis. The mesh contained 11,200 elements and utilized four-node, bilinear quadrilateral hybrid elements with constant pressure (ABAQUS type CAX4H). Boundary conditions were imposed such that nodes along the bottom of the substrate were constrained in the X and Y directions. FEM parameters were chosen from within bounds reported for mammalian tissues to generate displacement-force curves in close agreement with those observed in ex-vivo skin-nerve preparations. The resulting parameters were $C10 = 14,847$ and $C01 = 41,410$ for the Mooney-Rivlin skin model, and $E = 906098$ for the linear-elastic substrate. Prony parameters were $g_1 = 0.391$, $\tau_1 = 0.25$, $g_2 = 0.226$, and $\tau_2 = 9.371$. These parameters governed the model's transformation of indentation into SED, a measure of tissue distortion that correlates with the intensity of the SAI afferent response (*Dandekar et al., 2003*). SED was sampled from two elements approximating the volume and location of the SAI-afferent end organ, which was located beneath the cylindrical probe that contacted the model's surface. This probe was represented as a rigid analytic surface with a friction coefficient of 0.3 between the probe tip and skin. Due to the large diameter of the cylindrical probe (3 mm) relative to mouse touch domes (~0.1 mm), SED magnitude was assumed to be uniform across all Merkel cell–neurite complexes in the end organ.

#### Transduction functions

SED served as input to transduction functions representing clusters of Merkel cell–neurite complexes. Each transduction function is represented by *Equation 1*, where $I$ is current, $\varepsilon$ is SED, $\beta$ is an offset, $M$ is the number of Merkel cell–neurite complexes in the cluster, and $\alpha$ and $\lambda$ are gains for SED and the first derivative of SED, respectively. SED was converted to current with a sampling frequency of 1000 Hz for the model. The deterministic current was modified with the addition of a sample from the noise distribution, $\omega$. This noise distribution was a 7-point moving average of Gaussian noise with a mean of zero and a standard deviation set to reproduce variable inter-spike intervals characteristic of mouse SAI afferents (*Wellnitz et al., 2010*). Gaussian deviates were obtained using the Box–Muller method.

$$I(t) = \beta + M\left(\alpha\varepsilon(t) + \lambda\frac{d\varepsilon}{dt}\right) + \omega(t) \tag{1}$$

## Leaky integrate-and-fire models

Currents originating from transduction functions were input into an array of leaky integrate-and-fire models representing spike initiation zones. Neural dynamics were abstracted to a single differential equation (*Equation 2*), where *R* is resistance, *C* is capacitance, *u(t)* is membrane potential, and *I(t)* is current. In myelinated sensory afferents, a spike generated at one spike initiation zone antidromically invades the other spike initiation zones, resetting them and initiating an absolute refractory period (*Lindblom and Tapper, 1966*; *Horch et al., 1974*; *Fukami, 1980*; *Peng et al., 1999*). Thus, when a spike was generated by a given leaky integrate-and-fire model, that spike reset all other leaky integrate-and-fire models in the simulated end organ. When current drove the membrane potential to the spike initiation threshold, $\bar{v}$, a spike time was recorded and a 1-ms absolute refractory period was initiated. Numeric evaluation of the leaky integrate-and-fire equation was performed with the fourth order Runge–Kutta method.

$$RC\frac{du}{dt} = -u(t) + RI(t)$$  (2)

Note that computational models ignore the neurite lengths between Merkel cell–neurite complexes, spike initiation zones and branch point nodes. Calculations of current dynamics in neurites and myelinated branches justify this exclusion if three criteria are met: (1) neurites are electrotonically compact enough to efficiently spread receptor current to heminodes, (2) current spread from transduction units to spike initiation zones is faster than the inactivation time course of sodium channels, and (3) action potential spread through myelinated branches to the node where all branches converge is faster than the refractory period at nodes. Based on our morphometric data, these criteria are met.

First, the mean path length from neurite tips to heminodes is 33.7 μm (N = 32 neurites from three touch domes). Using published values for $R_m$ and $R_i$ (15,000 Ω·cm², 125 Ω·cm; *Baranauskas et al., 2013*) and estimated neurite diameters of 1 μm obtained from whole-mount imaging, the calculated length constant for these neurites is 548 μm, which exceeds neurite path lengths by an order of magnitude. Thus, we conclude that electrotonic spread in these neurites is likely to be efficient and no cable models are needed to predict the dynamics of receptor current in the neurites.

Second, based on conduction velocity values of unmyelinated fibers with similar diameters, we estimate conduction velocity in neurites to be 0.7 m/s (*Cain et al., 2001*). Current spread from the shortest path measured (14.3 μm) is estimated to be 0.02 ms, and from the longest (64.2 μm), 0.15 ms. Although these times vary by an order of magnitude, they are at least one order of magnitude lower than the inactivation duration of sodium channels (*Engel and Jonas, 2005*).

Third, we estimate a conduction velocity of 13 m/s in myelinated branches based on previous recordings from mouse SAI afferents (*Wellnitz et al., 2010*). To determine whether myelinated branches in the end-organ are thinner than branches in nerve trunks and consequently have lower conduction velocities, we measured diameters at different orders of branching in the end-organ. Branch thicknesses were not significantly different in the nerve trunk, primary and secondary branches of SAI afferents (N = 3 touch domes, ≥4 measurements per branch order). Based on these estimates, travel times along branches to the node where they converge for the longest (245.4 μm) and shortest (122.2 μm) paths are calculated to be 0.019 and 0.009 ms, respectively. These times are at least two orders of magnitude shorter than the estimated refractory period of 1 ms, and can be abstracted away in our model because differences in travel time are not sufficient to induce delays or competitive interactions at nodes of Ranvier.

## Model fitting

Fitting each end-organ model to the mean SAI afferent response involved three free parameters in the transduction function: *β*, *α*, and *λ*. These were selected with gradient free response surface methodology using Latin hyper-cube space filling designs, where each design was composed of 20 trial points (sampled using the LHS package in R). The start point was informed by a domain search utilizing 50 points in a space filling design. Skin-mechanics models were fitted as described above, and the leaky integrate-and-fire parameters were fixed at values of 5 ms, $1 \times 10^{-8}$ mF, and 30 mV for *τ*, *C*, and $\bar{v}$, respectively.

Each end-organ model configuration was fitted to a prototypical mouse SAI response. The prototypical SAI response was derived from linear regressions of ramp-phase and static-phase firing rates

recorded from mouse SAI afferents as described above (N = 4 units). Fits maximized the combined goodness of fit, measured as fractional sum of squares (*Equation 3*), between biological data and the model's simulated firing rates. This combined goodness of fit has a value of 2 for a model that perfectly matches the biological response profile. For stimulus $i$, $\overline{hfr_i}$ and $hfr_i$ represents the biologically observed and simulated static (hold) firing rate, respectively, and $\overline{rfr_i}$ and $rfr_i$ are the biologically observed and simulated dynamic (ramp) firing rate, respectively. The index $i$ spanned from 1 to 75 to include five displacement depths and three accelerations giving 15 unique stimulations, each of which was simulated five times for a given set of model parameters.

$$fss_{combined} = \left(1 - \frac{\sum_{i=1}^{75}\left(\overline{hfr_i} - hfr_i\right)^2}{\sum_{i=1}^{75}\left(\overline{hfr_i}\right)^2}\right) + \left(1 - \frac{\sum_{i=1}^{75}\left(\overline{rfr_i} - rfr_i\right)^2}{\sum_{i=1}^{75}\left(\overline{rfr_i}\right)^2}\right) \qquad (3)$$

After fitting, $\beta$ took the value of $5.643 \times 10^{-8}$, $5.648 \times 10^{-8}$, $5.669 \times 10^{-8}$, and $5.672 \times 10^{-8}$ mA for model configurations of {8, 5, 3, 1}, {7, 6, 4, 2, 1}, {6, 4, 3}, and {5, 4, 3, 1}, respectively. Values for $\alpha$ were $2.539 \times 10^{-14}$, $2.386 \times 10^{-14}$, $2.612 \times 10^{-14}$, and $2.641 \times 10^{-14}$ mA/Pa for {8, 5, 3, 1}, {7, 6, 4, 2, 1}, {6, 4, 3}, and {5, 4, 3, 1}. Finally, $\lambda$ values were $5.833 \times 10^{-11}$, $4.994 \times 10^{-11}$, $6.211 \times 10^{-11}$, and $6.491 \times 10^{-11}$ mA·ms/Pa. These values were used for the first two computational experiments (*Figures 4E and 5A*), where results were generated for each prototypical end organ. By contrast, model parameters for the two end-organ configurations in the third computational experiment (*Figure 5B*) were set as $\beta = 5.658 \times 10^{-8}$ mA, $\alpha = 2.545 \times 10^{-14}$ mA/Pa, and $\lambda = 5.882 \times 10^{-11}$ mA·ms/Pa, which represent the averages of all previous configuration parameters.

To compare firing rates of configurations in *Figure 5A* and *Table 1*, the change in firing rate was defined as the difference in summed firing rates across all stimulations divided by the lowest summed firing rate of the two configurations. This is described by *Equation 4*, where $fr_{a_i}$ and $fr_{b_i}$ are firing rates generated by the two configurations for stimulation $i$.

$$\% fr\, change = \frac{\sum_{i=1}^{75} fr_{ai} - \sum_{i=1}^{75} fr_{bi}}{\sum_{i=1}^{75} fr_{bi}}, \quad where \sum_{i=1}^{75} fr_{ai} > \sum_{i=1}^{75} fr_{bi} \qquad (4)$$

## Statistics

Statistical analyses were performed in Prism 5 (Graphpad Software, La Jolla, CA). Data were fitted either with linear regressions or single exponentials as indicated. Significant differences between best–fit curves were assessed by comparing $\kappa$ and $Y_0$ of the exponential fits with extra sum-of-squares F tests. The distribution of Merkel cells to terminal neurites was fitted with a Gaussian distribution ($R^2 > 0.99$).

## Acknowledgements

Thanks to Drs David Ginty and Michael Rutlin for sharing their whole-mount immunostaining protocol prior to publication, Drs Carol Mason and Punita Bhansali and Ms Austen Sitko for assistance with Neurolucida. Some experiments were performed in the Department of Neuroscience at Baylor College of Medicine.

---

## Additional information

### Funding

| Funder | Grant reference number | Author |
|---|---|---|
| National Institute of Neurological Disorders and Stroke | R01 NS073119 | Gregory J Gerling, Ellen A Lumpkin |
| National Institute of Arthritis and Musculoskeletal and Skin Diseases | P30 AR044535 | Ellen A Lumpkin |
| National Institute of General Medical Sciences | T32 GM007367 | Blair A Jenkins |

The funders had no role in study design, data collection and interpretation, or the decision to submit the work for publication.

## Author contributions

DRL, Computational modeling, Analysis and interpretation of data, Drafting or revising the article; KLM, Quantitative morphometeric analysis, Analysis and interpretation of data, Drafting or revising the article; SAW, Electrophysiology, Edited article, Analysis and interpretation of data; BAJ, Assisted with cryosection immunohistochemistry and morphometry, Edited article, Analysis and interpretation of data; YB, Electrophysiology, Analysis and interpretation of data; MNR, Contributed essential reagents, Edited article; GJG, EAL, Conception and design, Analysis and interpretation of data, Drafting or revising the article

## Ethics

Animal experimentation: Animal use was conducted according to guidelines from the National Institutes of Health's Guide for the Care and Use of Laboratory Animals and was approved by the Institutional Animal Care and Use Committees (IACUC) of Baylor College of Medicine (protocol #AN4160) and Columbia University Medical Center (protocol #AC-AAAC1561).

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
