## [Decision Letter]

Thank you for sending your work entitled “Computation identifies structural features predicted to govern neuronal firing properties in mammalian touch receptors” for consideration at *eLife*. Your article has been favorably evaluated by a Senior editor, a Reviewing editor, and 3 reviewers.

The Reviewing editor and the reviewers discussed their comments before we reached this decision, and the Reviewing editor has assembled the following comments to help you prepare a revised submission.

This study combines neuroanatomical reconstruction, electrophysiological recordings, and computational modeling to understand how neuronal architecture of branched mechanoreceptors (i.e., SAI afferents innervating Merkel cells) affects firing properties. The authors employed immunohistochemistry to identify heminodes (the initiation site of action potentials) and nodes of Ranvier of SAI afferents. They found that there was a large degree of variability in peripheral innervation and firing properties of SAI afferents. Strikingly, the difference in mechanical sensitivity of different SAI afferents does not directly correlate with the total number of innervated Merkel cells (i.e., afferents innervating a smaller number of Merkel cells can have a similar mechanical sensitivity to that of afferents innervating many more Merkel cells). Using a combination of computational models with consideration of skin compression, sensory transduction, and integration of afferent branching, they propose that the variability is likely due to an asymmetric distribution of Merck cell innervation by a single afferent with multiple branches. Their models predict that touch evoked firing is increased when Merkel cell-nerve complexes are arranged in a skewed distribution among heminodes, which provides an underlying mechanism for the driver effect of a dominant branch over other minor branches proposed by previous studies.

The manuscript thus represents a very nice “proof-of-concept” that firing properties of slowly adapting Type I mechanoreceptors (SAI) can be predicted mathematically. The authors have done a beautiful job in reconstructing the morphology of sensory end organs (Merkel cell-neurite associations) and their relationship to the location of spike initiation zones (hemi-nodes) in touch domes, and provided clear logic during the construction of their model.

In sum, the authors have provided important new information on the organization of Merkel-cell receptors using a combination of structural and functional data together with computer modeling, making a strong case for the impact of structural features on afferent firing patterns. Although the quantitative aspects relate specifically to Merkel-cell receptors, the general conclusions are likely to be equally applicable to other branched afferents with multiple encoding sites. Our principal concern relates to the first “main issue” noted below.

Main issues:

1) The authors used reconstructions of Merkel cell-neurite arbors from touch domes innervated by a single myelinated fiber to guide configuration of their model so that it could accurately predict “prototypical” responses obtained by recordings from SAI fibers. The underpinning assumption is that the morphological relationships of Merkel cell-neurite associations and hemi-nodes in individual touch domes form a valid starting point for mathematical modeling of the firing properties recorded from SAI afferent fibers. However, while most touch domes are innervated by a single large myelinated axon, the converse is not true, as the majority of SAI afferent fibers innervate multiple touch domes (as detailed in the cited literature). Importantly, because the authors used an extremely large (3 mm diameter) probe tip to apply stimuli and because touch domes are closely spaced (<1 mm apart, cited by authors), their large probe tip certainly stimulated multiple touch domes, up to 5 (and potentially more). There is a high probability, therefore, that the single fibers recorded by the authors innervated multiple touch domes, and thus the electrophysiological records that are pivotal to the validation of their model, likely represent the combined output from multiple touch domes stimulated simultaneously. These separate outputs would be expected to interfere substantially at branch points on the parent axon distal to the recording electrode, which could explain the wide variation observed by the authors in firing rate across fibers. In addition, without knowing how many touch domes contributed to the responses recorded from single SAI fibers, conclusions that mechanical sensitivity of SAI fibers does “not scale with total Merkel cell number” are questionable since total Merkel cell number is not known. Although it is indeed interesting that the authors were able to model actual recordings, uncertainty remains whether this ability might attest more to the flexibility (i.e., reconfigurability) of the model than to its biological validity.

This major concern – that the authors do not know how many touch domes contributed to recorded responses due to the large size of their probe – could potentially be rectified (or partially rectified) by performing electrophysiological recordings using a smaller probe in *Atoh1/nGFP* mice where visual feedback can be used to ensure that recorded responses represent the output from a single touch dome. Of course it would be even more valuable if the same individual touch domes were also reconstructed morphologically for subsequent modeling purposes, as discussed by the authors.

The one piece of data that appears to address this issue is in Figure 3 A–C. The authors state that the dotted line in Figure 3 is the receptive field of the SAI afferent whose responses are given in Figure 3. The receptive field size is therefore much smaller than the probe tip used for mechanical stimulation, and the important thing is that there is only one SAI afferent (this one) that was stimulated. We presume that the authors must have been able to identify single touch domes with receptive fields in this way in order to obtain the data plotted in Figure 3. We also presume that if there had been additional touch domes innervated by the same afferent, they must have been able to detect that. We think that this is the case based on the description of the properties of SAI afferents given by some of the authors in their J Neurophysiol paper ([60]; 103, 3378), which they cite for details of the electrophysiological method (see especially page 3384 where they mention recording from 2 afferents each innervating 2 touch domes).

As should be clear from the preceding paragraphs, it would be most helpful if the authors could clarify this issue and consider providing additional experimental data.

2) Related to point #1, because of collision/antidromic “resetting” at branch points on the parent axon, the dome that produces the first action potentials to arrive at the branch will dominate the recordings (as shown by Lindblom and Tapper in 1969; Integration of impulse activity in a peripheral sensory unit. Expl Neurol. 15, 63-69). Thus, if two or more domes are activated simultaneously, the spikes recorded more proximally along the parent axon will generally represent the output from one dome only. However, which dome dominates the response is unclear when multiple domes are stimulated simultaneously. If both domes are stimulated simultaneously, you generally get one or the other pattern, and which pattern you get depends on which dome is closer to the branch point (or conducts fastest to reach the branch point first). This effect makes it difficult to connect the anatomical reconstructions with the physiology if one doesn't know which dome provided the output they recorded. The authors should comment on this issue.

3) The driver effect predicts that specific stimulation of partial receptive field innervated by a dominant branch should generate similar responses as those evoked by whole receptive field stimulation. Does the model conform to this prediction?

According to the data and models, travel times along branches to the node are much shorter than that of antidromic resetting. It would be interesting to examine whether the initial firings, which should not be affected by the resetting mechanism, from SAI afferents with more Merkel endings are always stronger than those from afferents with fewer Merkel endings no matter how Merkel endings are distributed among different heminodes. This information can be obtained by re-examining existing recording data and the simulations.

---

## [Author Response]

*1) […] There is a high probability, therefore, that the single fibers recorded by the authors innervated multiple touch domes, and thus the electrophysiological records that are pivotal to the validation of their model, likely represent the combined output from multiple touch domes stimulated simultaneously*.

The reviewers noted that some SAI afferents innervate multiple touch domes and that the 3-mm probe tip used to gather the electrophysiology data in this study is large enough to stimulate multiple touch domes simultaneously. To address this issue, before applying controlled indentations using the 3-mm probe tip, we manually probed the skin surface with a mechanical stimulator whose tip was smaller than an individual touch dome (100 µm) to specifically test how many touch domes were innervated by the recorded SAI afferent. For computational modeling, we only included fibers that responded to stimulation of a single touch dome. Thus, we are certain that only the Merkel cells in that touch dome could be contacted by the afferent. We thank the reviewers for noting this oversight in the text and we have clarified our procedure accordingly.

Based on published data from cats and neonatal mice, the reviewers raised the concern that most SAI afferents innervate multiple touch domes. To directly address this concern, we analyzed a larger dataset of SAI afferent recordings from adult mouse hindlimb, which is the recording site used in this study. In the mouse strain used here (*Atoh1/nGFP* transgenic mice on a BDF1 strain background), we found that the majority of SAI afferents innervate a single touch dome (70%; N=27 SAI afferents). We feel that this analysis is a valuable addition to the literature and have included the results in the revised submission.

*2) Related to point #1, because of collision/antidromic “resetting” at branch points on the parent axon, the dome that produces the first action potentials to arrive at the branch will dominate the recordings (as shown by Lindblom and Tapper in 1969; Integration of impulse activity in a peripheral sensory unit. Expl Neurol. 15, 63-69). Thus, if two or more domes are activated simultaneously, the spikes recorded more proximally along the parent axon will generally represent the output from one dome only. However, which dome dominates the response is unclear when multiple domes are stimulated simultaneously. If both domes are stimulated simultaneously, you generally get one or the other pattern, and which pattern you get depends on which dome is closer to the branch point (or conducts fastest to reach the branch point first). This effect makes it difficult to connect the anatomical reconstructions with the physiology if one doesn't know which dome provided the output they recorded. The authors should comment on this issue*.

We agree that an SAI afferent with more than one receptive field will have interesting firing properties that will differ depending on the receptive field stimulated. These possibilities are not addressed in the present manuscript because, as mentioned above, the electrophysiology data presented here are from SAI afferents that only innervate individual touch domes. Thus, we sought to define the fundamental firing properties set by structural components of a single receptive field. The implications of having multiple punctate receptive fields, as described by Lindblom and Tapper, are now discussed.

*3) The driver effect predicts that specific stimulation of partial receptive field innervated by a dominant branch should generate similar responses as those evoked by whole receptive field stimulation. Does the model conform to this prediction*?

Our model recapitulates the driver effect because it incorporates resetting of all spike initiation zones in the arbor when an action potential fires from one branch. This is a key component of the driver effect because it suppresses firing from the other branches, which gives a bias to the spike initiation zone that fires first. Although noise in transduction models introduces an element of stochasticity, the branch with the most transduction units has the highest likelihood of reaching spike threshold first.

Given this architecture, the reviewers’ prediction should be upheld by the computational model if one could deliver the same state of stress to just the driver branch as is delivered to the entire touch dome. We cannot confirm this quantitatively with our present models, which are built to simulate flat-field rather than punctate stimuli. During these skin-nerve electrophysiological recordings, we use a 3-mm diameter, filleted stimulus probe to deliver a consistent state of stress to the entire receptive field. This helps accommodate for small but inevitable imprecision in stimulus placement and avoids the uncontrolled stress gradients that would occur with a spherical stimulus. To computationally represent this flat-field stimulus, we have modeled the skin with elements that have edge lengths of 100 µm. This makes it impossible to apply stress to only the driver branch because it would occur on a sub-element scale. To model a punctate stimulus across different branches will require building, validating and experimentally constraining new finite element models with a finer discretized mesh. Future studies will address this fascinating question.

*According to the data and models, travel times along branches to the node are much shorter than that of antidromic resetting. It would be interesting to examine whether the initial firings, which should not be affected by the resetting mechanism, from SAI afferents with more Merkel endings are always stronger than those from afferents with fewer Merkel endings no matter how Merkel endings are distributed among different heminodes. This information can be obtained by re-examining existing recording data and the simulations*.

We agree that this is an interesting analysis. We re-analyzed our SAI afferent recording data to test for a relationship between the latency of first spikes, which will not be impacted by zone resetting, and the total number of Merkel cells. We focused on suprathreshold stimulus magnitudes that reliably elicited sustained firing.

Although we directly measured the number of Merkel cells in each touch dome for recorded SAI afferents, our reconstructions suggest that up to 15% are not contacted by the afferent (see Figure 1). To account for this, we grouped small touch domes (12 and 13 Merkel cells) and large touch domes (20 and 22 Merkel cells) to represent our level of resolution. Consistent with the reviewers’ prediction, first spike latencies were significantly shorter in the large touch dome group (mean±SEM, *N*; for large touch domes: 10.9±1.6 ms, N=57; for small touch domes: 40.0±14.5 ms, *N*=60; P=0.027; Student’s unpaired *t* test, one-tailed). We also noted that the variability of first spike latencies was significantly higher in the small touch dome group (P<0.0001), which suggests that touch domes with fewer transduction units fire less reliably during dynamic stimuli.